# New Approaches in the Management of Sudden Cardiac Death in Patients with Heart Failure—Targeting the Sympathetic Nervous System

**DOI:** 10.3390/ijms20102430

**Published:** 2019-05-16

**Authors:** Márcio Galindo Kiuchi, Janis Marc Nolde, Humberto Villacorta, Revathy Carnagarin, Justine Joy Su-Yin Chan, Leslie Marisol Lugo-Gavidia, Jan K. Ho, Vance B. Matthews, Girish Dwivedi, Markus P. Schlaich

**Affiliations:** 1Dobney Hypertension Cenre, School of Medicine—Royal Perth Hospital Unit, Faculty of Medicine, Dentistry & Health Sciences, The University of Western Australia Level 3, MRF Building, Rear 50 Murray St, Perth 6000, MDBP: M570, Australia; marcio.galindokiuchi@uwa.edu.au (M.G.K.); janis.nolde@uwa.edu.au (J.M.N.); revathy.carnagarin@uwa.edu.au (R.C.); justine.chan@uwa.edu.au (J.J.S.-Y.C.); 22532229@student.uwa.edu.au (L.M.L.-G.); jan.ho@uwa.edu.au (J.K.H.); vance.matthews@uwa.edu.au (V.B.M.); 2Cardiology Division, Department of Medicine, Universidade Federal Fluminense, Niterói, Rio de Janeiro 24033-900, Brazil; huvillacorta@globo.com; 3Harry Perkins Institute of Medical Research and Fiona Stanley Hospital, The University of Western Australia, Perth 6150, Australia; girish.dwivedi@perkins.uwa.edu.au; 4Departments of Cardiology and Nephrology, Royal Perth Hospital, Perth 6000, Australia; 5Neurovascular Hypertension & Kidney Disease Laboratory, Baker Heart and Diabetes Institute, Melbourne 3004, Australia

**Keywords:** heart failure, positron emission tomography, renal denervation, sudden cardiac death, sympathetic nervous system, ventricular arrhythmias, hypertension

## Abstract

Cardiovascular diseases (CVDs) have been considered the most predominant cause of death and one of the most critical public health issues worldwide. In the past two decades, cardiovascular (CV) mortality has declined in high-income countries owing to preventive measures that resulted in the reduced burden of coronary artery disease (CAD) and heart failure (HF). In spite of these promising results, CVDs are responsible for ~17 million deaths per year globally with ~25% of these attributable to sudden cardiac death (SCD). Pre-clinical data demonstrated that renal denervation (RDN) decreases sympathetic activation as evaluated by decreased renal catecholamine concentrations. RDN is successful in reducing ventricular arrhythmias (VAs) triggering and its outcome was not found inferior to metoprolol in rat myocardial infarction model. Registry clinical data also suggest an advantageous effect of RDN to prevent VAs in HF patients and electrical storm. An in-depth investigation of how RDN, a minimally invasive and safe method, reduces the burden of HF is urgently needed. Myocardial systolic dysfunction is correlated to neuro-hormonal overactivity as a compensatory mechanism to keep cardiac output in the face of declining cardiac function. Sympathetic nervous system (SNS) overactivity is supported by a rise in plasma noradrenaline (NA) and adrenaline levels, raised central sympathetic outflow, and increased organ-specific spillover of NA into plasma. Cardiac NA spillover in untreated HF individuals can reach ~50-fold higher levels compared to those of healthy individuals under maximal exercise conditions. Increased sympathetic outflow to the renal vascular bed can contribute to the anomalies of renal function commonly associated with HF and feed into a vicious cycle of elevated BP, the progression of renal disease and worsening HF. Increased sympathetic activity, amongst other factors, contribute to the progress of cardiac arrhythmias, which can lead to SCD due to sustained ventricular tachycardia. Targeted therapies to avoid these detrimental consequences comprise antiarrhythmic drugs, surgical resection, endocardial catheter ablation and use of the implantable electronic cardiac devices. Analogous NA agents have been reported for single photon-emission-computed-tomography (SPECT) scans usage, specially the ^123^I-metaiodobenzylguanidine (^123^I-MIBG). Currently, HF prognosis assessment has been improved by this tool. Nevertheless, this radiotracer is costly, which makes the use of this diagnostic method limited. Comparatively, positron-emission-tomography (PET) overshadows SPECT imaging, because of its increased spatial definition and broader reckonable methodologies. Numerous ANS radiotracers have been created for cardiac PET imaging. However, so far, [^11^C]-meta-hydroxyephedrine (HED) has been the most significant PET radiotracer used in the clinical scenario. Growing data has shown the usefulness of [^11^C]-HED in important clinical situations, such as predicting lethal arrhythmias, SCD, and all-cause of mortality in reduced ejection fraction HF patients. In this article, we discussed the role and relevance of novel tools targeting the SNS, such as the [^11^C]-HED PET cardiac imaging and RDN to manage patients under of SCD risk.

## 1. Introduction

### 1.1. Epidemiology of Sudden Cardiac Death

Cardiovascular diseases (CVDs) have been considered the most predominant causes of death and one of the most critical public health issues worldwide [1]. In 2011, the United Nations officially defined non-communicable diseases, comprising CVDs, as the primary interest for global health and set out a strategic plan to significantly decrease the potential adverse consequences of these conditions [2]. The greater attention spend on these universal non-communicable disease aims has helped to trigger efforts to follow and benchmark national hard work at reducing CVD and further non-communicable conditions [3,4].

Over the last two decades, cardiovascular (CV) mortality has reduced in high-income countries [5] due to the implementation of protective actions to decrease the impact of coronary artery disease (CAD) and heart failure (HF). In spite of these promising outcomes, CVDs provoke ~17 million deaths per year globally of which ~25% are due to sudden cardiac death (SCD) [6]. In Europe, deaths from SCD affect roughly 700,000 people per year [7,8,9,10,11]. Also, in the US, around one million deaths occur annually from CVDs, of which 330,000 are attributed to sudden death [12,13]. A clear correlation has been demonstrated between structural cardiac diseases and SCD. In >70% of patients, the causal cardiac condition is myocardial ischemia [14]. SCD risk is more significant in male than in female patients, and the risk growths according to age, mainly due to the increased prevalence of CAD in the aged population [15]. It is expected that the frequency of SCD varies between 1.4 and 6.7/100,000 individual-years [15]. Recent recommendations for managing ventricular arrhythmias (VAs) report that SCD in the young has an expected incidence of 0.46–3.7 episodes per 100,000 individual years [16,17,18], equivalent to a ~1,100–9,000 deaths in Europe, and from 800 to 6200 deaths every year in the US [19].

### 1.2. Treatment Indication

For more than 20 years, researchers have proposed various “indicators” for SCD due to their association with cardiac ischemic disease. Among them are non-invasive SCD risk indicators, which might prove helpful in cases of myocardial ischemia [20]. On the other hand, in spite of the encouraging early studies, none of these “prediction parameters” changed in the real world our daily clinical management of patients. Indeed, left ventricular ejection fraction (LVEF) is the single predictor that has reliably presented a relationship with higher risk of SCD in subjects presenting with myocardial infarction (MI) and left ventricular (LV) dysfunction [21,22]. LVEF has been in use for >10 years targeting the benefit of an automatic implantable cardioverter-defibrillator (ICD) for primary prevention of SCD, usually combined with the New York Heart Association (NYHA) functional class. Despite the fact that LVEF is considered an inaccurate clinical parameter, it is still primarily utilized to indicate patients for ICD implantation in the SCD primary prevention. The ICD has been extremely beneficial in the secondary prevention of SCD; subjects with previous MI and severe systolic LV impairment [21,23], survivors of cardiac arrest or in those with sustained ventricular tachycardia (VT) at high risk of new episodes [24]. Current treatments consist of antiarrhythmic medications, surgical septal myectomy (resection of the ventricular septal wall) in individuals presenting hypertrophic cardiomyopathy and LV outflow region obstructions, endocardial ventricular catheter-ablation and implant of electronic cardiac devices [14].

### 1.3. Prognosis

HF prognosis is usually unfavorable. In patients being admitted with decompensated HF to hospital, the one-year mortality rate is ~20% in persons <75 years old and higher than 40% in those >75 years, despite current clinical treatments [25,26]. High-grade data about the projection of outpatient HF populaces is challenging to characterize. Subjects in clinical trials tend to be younger and with healthier in comparison to daily clinical practice and therefore have a superior prognosis, with a twelve-monthly mortality rate of 5–10% in contemporary studies, although trial designs prevented very-low-risk subjects to get enrolled [27,28]. On the other hand, treatment seems to have had an extraordinarily positive effect on the outcome of chronic HF patients across the previous two decades. For instance, in the HEart Failure trial V-HeFT-I the median life expectation of participants was only 3.5 years in comparison to >8 years for a similar age group with moderate to critical HF on treatment with medication and additional cardiac resynchronization therapy (CRT), as suggested by the CARE-HF (CArdiac REsynchronization in Heart Failure) [29,30,31,32].

### 1.4. Pathophysiology of HF and Sympathetic Nerve Activity Synonym

Myocardial systolic functional impairment is related to neuro-hormonal overactivity as a compensating system to preserve cardiac output when heart function is deteriorating. The neuronal role of this reaction is characterized using an increased sympathetic nervous system (SNS) cardiac traffic [33]. The most critical areas of the brain that synonym may be associated with chronic HF are the rostral ventrolateral medulla (RVLM) and the paraventricular nucleus (PVN). Therefore, chronic HF may be understood as illness of the central nervous system as it is correlated to autonomic dysregulation with a removal of parasympathetic activity and a higher sympathetic nerve fire rates [34,35]. In the case of reduced systolic function, the sympathetic activation might reproduce the net balance and the relations between proper reflex compensatory responses to deteriorated systolic function, and excitatory stimuli that prompt catecholaminergic reactions in excess of homeostatic necessities. Anomalous SNS triggering provoked by the boost of excitatory inputs has been described (e.g., alterations in reflexes of peripheral baro- and chemo-receptors, in the sympathetic outflow controlled by chemical mediators, and in the sympathetic activation processed by central systems) [36,37]. The hormonal component is represented by increased secretion and elevated levels of some endocrine parameters, predominantly adrenaline (A) and noradrenaline (NA), as well as, the factors of the RAAS (i.e., renin, angiotensin II and aldosterone) [38]. SNS overactivity is evinced by augmented plasma NA and A levels, raised sympathetic discharge of the CNS, and enhanced NA spillover [39]. Roughly 50-fold Cardiac NA spillover levels can be measured in HF subjects without treatment compared to those of healthy individuals under vigorous exercise circumstances [40]. Patients suffering from end stage systolic HF can present reduced sympathetic neuronal density and function, leading to declined cardiac NA levels, as well as diminished postsynaptic β-adreno-receptor (AR) density, because of the exhaustion of cardiac SNS neuronal NA stores and reduced NA presynaptic reuptake secondary to NA-transporter down-regulation [41,42].

The harmfulness of the SNS on the heart is well established. For example, intravenous injection of isoproterenol (non-selective β-AR agonist) or NA lead to acute contraction band lesions attributed to relative oxigen deprivation, calcium overload, the raising of cyclic adenosine monophosphate (cAMP), excitation of α- and β-ARs, and generation of reactive oxygen species [43,44]. Chronic catecholaminergic excitation can be a trigger of heart fibrosis, diminish adrenergic and inotropic reserves, and prompt cardiac apoptosis and dysfunction through LV dilatation [44,45]. Furthermore, NA promotes cardiac cell death by means of both β_1_AR-mediated and reactive oxygen species/tumor necrosis factor/caspase-mediated signaling pathways [46,47]. An interesting illustration of catecholamine-induced heart damage occurs in stress mediated cardiomyopathy (Takotsubo syndrome); excessive circulating levels of adrenaline elicit a form of cardiac stunning on a cellular level that comprises the signaling switch of the cardiac β_2_-AR from Gs to Gi proteins, particularly in the apical myocardium where β-AR density is highest [48], consequently impairing inotropy [49]. Kuniyoshi and colleagues showed that in subjects with advanced HF and critical systolic functional impairment, CRT implantation lead to a substantial decrease of muscle sympathetic nerve activity (MSNA) at rest and during handgrip exercise, in contrast to HF patients receiving only medical therapy (Figure 1). Furthermore, MSNA decreases after CRT had a negative association with O_2_ expenditure [50]. Previously, Grassi and colleagues also reported that MSNA at rest reduced two months post-CRT implantation. Remarkably, plasma NA levels did not fall [51].

The pathophysiology of SCD is clearly multifactorial and it is thought to involve the interface between a transitory event and a pre-existing substrate, which leads to electrical instability and VAs, leading to failure of circulation. Comprehending the processes that provoke these proceedings may benefit us to elucidate when the interaction between an eliciting event and a present substrate can become dangerous. Anatomical and physiological remodeling of the heart, fibrosis and calcification of vessels, autonomic imbalance, volume and electrolyte disturbances are considered to be relevant contributors predisposing HF patients to SCD. Structural changes can modify myocardial electrophysiological properties. Fibrotic processes on a cellular level interrupt the regular structure and causes a diminution in conduction speed via the unhealthy cardiac tissue [52]. This disorder may create heterogeneous zones of conveyance and activation, delaying ventricular depolarization and producing late potentials in the end-segment of QRS complexes, which can sustain reentrant arrhythmias, such as VT [53,54,55].

### 1.5. Relevant Diagnostic Approaches: PET Scan and Cardiac SNS Imaging

Given the relevance of the SNS in the context of SCD, imaging modalities to assess sympathetic function seem a plausible approach to patients at risk. Analogous NA agents have been reported for single photon-emission-computed-tomography (SPECT) scans usage, specially the ^123^I-metaiodobenzylguanidine (^123^I-MIBG) [56,57,58]. Currently, HF prognosis assessment has been improved by this tool. Nevertheless, this radiotracer is costly, which makes the use of this diagnostic method limited. Comparatively, positron-emission-tomography (PET) overshadows SPECT imaging, because of its increased spatial definition and broader reckonable methodologies. Furthermore, there are more autonomic nervous system (ANS) radiotracers for PET than for SPECT imaging. Hence, it allows us to a widespread assessment of cardiac ANS function.

Numerous ANS radiotracers have been created for cardiac PET imaging. However, so far, [^11^C]-meta-hydroxyephedrine (HED) has been the most significant PET radiotracer used in the clinical scenario. Growing data has shown the usefulness of [^11^C]-HED in important clinical situations, such as predicting lethal arrhythmias, SCD, and all causes of mortality in reduced LVEF HF patients [59].

Stimulatory cardiac effects (e.g., increased chrono-, dromo-, bathmo-, and inotropism) are mostly due to SNS triggering. The cardiac sympathetic signaling derives from the spinal cord preganglionic neurons on vertebral levels T1–T5 of the intermediolateral horn, linking with postganglionic dendrites in the interior of the sympathetic nerve network. Cardiac cervical and thoracic postganglionic neurons arrive at the cardiac plexus and originate sympathetic efferences that innervate atrial and ventricular chambers, the conveyance system, and coronary arteries [56]. Postganglionic SNS nerve terminations discharge stored NA from secretory vesicles after an action potential and nerve depolarization for subsequent binding to adrenergic, G protein-coupled receptors on effector cells. They are classified into the subtypes α and β. Subtypes α1, β1, and β2 of adrenergic receptors are the most common variants in the CV system [56]. Post-release, most of the NE in the synaptic cleft (50–80%) is reentering the presynaptic nerve-end through the NE reuptake transporter [60,61] (NET or uptake-1) in an energy consuming process. The greater part of the reabsorbed NE is packed into vesicles via the vesicular monoamine transporter (VMAT), whereas a minor portion is processed by catechol-O-methyltransferase and monoamine oxidase (MAO) (COMT) [56]. The residual NE is subject to reuptake in postsynaptic cells by the energy consuming, uptake-2 mechanism, or by diffusion into the vascular system (Figure 2). Existing SNS PET imaging radiotracers for clinical and experimental setting principally target postsynaptic adrenergic receptor density and presynaptic neural activity (e.g., uptake-1 and metabolism).

As previously mentioned, LVEF is considered the single parameter for recognizing subjects with an elevated SCD-risk who benefit from ICD implantation. However, the PAREPET (Prediction of ARrhythmic Events with Positron Emission Tomography) study proposed that quantifying heterogeneity in the sympathetic innervation of the heart could identify high-risk subjects for SCD [59]. They prospectively recruited 204 individuals eligible for primary prevention with ICDs suffering from ischemic cardiomyopathy with an LVEF of less than 35%. PET was utilized to measure the loss of myocardial sympathetic innervation (^11^C-meta-hydroxyephedrine [^11^C-HED]), viability (insulin-stimulated ^18^F-deoxyglucose) and perfusion (^13^N ammonia). The central goal was to find out if imaging of hibernating and/or denervated myocardium could be used as a predictive model for arrhythmic death in ischemic cardiomyopathy [59].

Figure 3 shows PET illustrations from two different subjects matching myocardial resting flow, viability, and sympathetic innervation, through the use of ^13^NH_3_, ^18^FDG and ^11^C-HED radiotracers, respectively. Quantitative image investigation concluded that the mean infarct volume was 20 ± 9 % of the LV. Comparatively, a much larger proportion of the LV tissue lost its innervation ~27% (*p *< 0.001 vs. infarcted), as well as, ~8% of the LV was denervated but kept its viability. In addition, infrequent LV hibernating myocardium (~3%) was reported, indicating a great importance of prior revascularization, and averaged [59].

The cumulative event rate curves of tertiles for each PET parameter are presented in Figure 4. The proportion of denervated myocardium had the most significant association with sudden cardiac arrest (SCA) (*p* = 0.001) (Table 1). The top, mid and bottom tertiles of denervation presented a ~6.7%, ~2.2%, and ~1.2%/year SCA event rates, respectively. A rise of 1% in the volume of denervated myocardium was estimated to lead to a 5.7% increase in SCA risk [59]. Moreover, an association was found in between the time elapsed until SCA and the volume of viable denervated myocardium (*p* = 0.025). The significant predictors of time to SCA occurrence are displayed in the Table 1 [59].

Individuals presenting a likely hazard of cardiac events have been identified by radiotracers generated from NET substrates currently used for cardiac imaging purposes. A novel ^18^F-labeled NET substrate—N-[3-Bromo-4-(3-[18F] fluoro-propoxy)-benzyl]-guanidine (LMI1195)—has been used experimentally, targeting to acquire better cardiac neuronal images through PET scans. It only has been possible because such agent has demonstrated a superior performance (e.g., exceptional cardiac uptake, heart-to-adjacent organ uptake ratios, and NET selectivity). This radiotracer also pursues particular features, which allows achieving the improved sensitivity, resolution, and quantification of PET images, revealing variations in the cardiac sympathetic innervation correlated to HF, that are crucial for possible treatment stratification [62].

## 2. Treatment Approaches Targeting the SNS: Renal Denervation

Renal innervation allows bidirectional interaction between the central nervous system and kidneys. Central and peripheral nervous system inputs modify efferent renal sympathetic nerve activity, which re-shapes the renal apparatuses (e.g., vessels, glomeruli, and tubuli) structurally and functionally [63]. Hence, renal blood circulation, glomerular filtration rate, tubular Na^+^ and H_2_O management suffer their influence, triggering the renin excretion from the juxtaglomerular apparatus which in turn controls part of the RAAS blood pressure (BP) and renal perfusion [63,64]. All of these factors have a significant role in the pathogenesis of chronic conditions such as hypertension, renal disease and HF [64]. This efferent signaling is furthermore determined by afferent activity from renal chemo- and mechano-receptors. The afferent innervation of the kidney follows the sympathetic nerves at kidneys level and then reaches the spinal cord though dorsal roots and finally brainstem areas involved in CV control [64,65,66,67]. Hence, the kidney denotes a basis of augmented sympathetic activity in the presence of some pathophysiological settings, such as renal ischaemia, hypoxia, and intrinsic renal disease [68,69,70]. The innervation of proximal and ventral segments of the kidney is denser in terms of peri-arterial sympathetic fibers, while distal and dorsal parts of the kidney are reached by fewer fibers of this kind. There is an evident great amount of efferent nerve fibers, with reducing density of afferent nerves from proximal to distal peri-arterial and renal parenchyma [65].

These notions can be regarded as supportive for the reasoning behind modulating autonomic innervation of blood vessels and other functional structures of the kidney, in order to decrease renal norepinephrine spillover and have a positive effect on pathophysiologies dependent on the sympathetic nervous system [71,72,73]. Some studies suggest that the efferent renal nerves are crucial to the renal hypoperfusion in chronic HF (CHF) [74]. CHF expressively reduces renal blood flow to innervated but not denervated kidneys by rising renal vascular resistance. These observations are indicative of the central role renal nerves play in the development of renal hypoperfusion in CHF. Also, acute renal denervation in rodents boosts renal blood flow in rats with CHF but not in control cases [75,76], leading us to believe that renal nerves have a tonic vasoconstrictive role in CHF. Relevantly, Kon et al. reported that acute renal denervation (RDN) leads to a reduction in glomerular capillary pressure while rising glomerular filtration rate in CHF, presenting evidence that the raised efferent renal sympathetic nerve activity in CHF damages renal function [77]. RDN markedly prevents progressive increases in BP and correlated renal damage and cardiac remodeling in hypertensive animal models [71,72]. Removal of sympathetic nerve activity to kidneys and following modifications in fluid mobilization and decreased levels of angiotensin II, as well as withdrawal of raised afferent renal nerve activity as a result of a pathophysiological change in kidneys, may play a role in RDN and its antihypertensive effects.

One study has revealed that RDN reduces but does not normalize the augmented plasma renin activity in experimental CHF [78], suggesting that the renal nerves are partially accountable for the maladaptive activity of the systemic renin–angiotensin system (RAS). Thus, efferent renal sympathetic nerve activity may influence the renal hypoperfusion, volume dysregulation and RAS activation in CHF. Interestingly, selective damage of afferent renal nerves had limited antihypertensive effects in hypertensive Dahl salt-sensitive rats [79,80]. Also, amplified sympathetic traffic is proposed to contribute to the progression of cardiac arrhythmias [81]. At the cellular level, NA is released from postganglionic neurons in reaction to sympathetic stimuli activating beta-receptors in the heart. As a consequence, changed cardiac calcium handling and electrophysiology participate in arrhythmogenic mechanisms, such as delayed after depolarization-related ectopic firing and re-entry [82,83].

The majority of brainstem areas participating in CV regulation receive input from afferent fibers of the kidney. Sympathetic afferent activations coming from the kidneys are capable of controlling nerve activity of numerous ganglia, including those that innervate the heart. For instance, there is good evidence for connections of renal sympathetic nerve activity and left stellate ganglion (LSG) activity [84,85]. The upregulation of LSG nerve growth factor expression and LSG neuronal activity have been related to renal sympathetic nerve stimulation [85]. LSG stimulation either results in the efferent sympathetic nerves triggering or boosted activation of the renal afferent fibers signaling in the direction of the brain. In addition, some studies assume that renal hemodynamic and excretory function seem to be affected by the frequency of stimulation of the renal nerves, but it has not been adequately confirmed. In canines, bilateral RDN has demonstrated substantial central and peripheral sympathetic nerve remodeling, enhanced baroreflex sensitivity and lowered stellate ganglion nerve activity [85]. RDN has been described as useful in modulating levels of catecholamines and overall body sympathetic nerve activity [85]. Fascinating, cardiac sympathetic hyperactivity measured by ^123^I-MIBG scintigraphy is markedly blunted post-selective RDN [86,87].

Brandt et al. revealed that besides BP-lowering effect, RDN as well substantially decreased LV mass and enhanced diastolic function as demonstrated by echocardiography. This might have weighty effects for the prognosis in resistant hypertensive subjects at elevated CV risk [88]. Afterwards, seventy-two refractory hypertensive patients were submitted to cardiac-MRI pre- and 6 months post-RDN (55 patients underwent RDN, and 17 assisted as controls [89]. RDN expressively decreased systolic and diastolic BP (~22/8 mmHg) with a decline of the indexed left ventricular mass (ILVM) (~7.1%), which were unaffected in the control group. The LVEF was boosted in individual post-RDN (43% vs. 50%; *p* < 0.001). In another study, Brandt et al. demonstrated that RDN improved E/E’ measurements of diastolic HF in patients [88]. In 15 out of 18 (83%) RDN non-responders subjects (response defined as systolic BP-lowering effect l ≥ 10 mmHg), the was expressively decreased. surprisingly, structural and functional cardiac modifications were in part independent of BP, indicating a direct modulating influence of the SNS activity.

Another study assessed 23 subjects with refractory hypertension who underwent cardiac-MRI and RDN [90]. ILVM, the extracellular volume fraction and indexed absolute extracellular volume were measured. RDN significantly reduced LV mass, whilst the extracellular volume continued unchanging, proposing that the perceived reduction in ILVM was not exclusively owing to a reverse of myocyte hypertrophy, but likewise owing to additional decline in collagen content. This points to myocardial interstitial fibrosis occurring in first instances. In 2005, Perlini et al. showed that sympathectomy or alpha-adrenergic blockade improved myocardial interstitial fibrosis secondary to hypertension in rodents [91].

In 2015, McLellan et al. [92] assessed 24-h ABPM, echocardiogram, cardiac-MRI and electrophysiological-study of 14 refractory hypertensive patients before and six months after RDN. The electrophysiological procedure comprised measurements of conduction times, effective refractory periods, and P-wave duration, which are markers in the development of atrial fibrillation. After RDN, the mean 24-h ABPM was decreased, whilst total conduction velocity notably rose, and conduction time reduced, supporting an improvement. A positive association between changes in conveyance velocity and modifications in mean 24-h ABPM was also reported. Pre-clinical data demonstrated that atrial remodeling happens at different time domains in chronic hypertension with substantial electrostructural association of the remodeling cascade. Primary establishment of antihypertensive therapy may avoid substrates development responsible for keeping AF [93]. Furthermore, a marked decrease in LV mass and spread ventricular fibrosis observed in CMR was observed [92], in keeping with previously mentioned studies above. Similarly, Dorr et al. [94] studied 100 consecutive refractory hypertensive patients who undergone RDN. Therapeutic response was well-defined as office systolic BP-lowering effect >10mmHg, six months post-procedure [95,96]. Blood tests for amino-terminal pro-peptide (PINP, PIIINP) and carboxyl-terminal pro-peptide (PICP) were performed pre- and 6 months post-RDN. The influence of RDN in increasing collagen absorption was assessed, as demonstrated by measurement of these particular biomarkers for reabsorption of cardiac extracellular matrix and CV fibrosis. A substantial fall of systolic BP (-24.3 mmHg), all pro-peptides serum levels were reported six months post-RDN in patients, supporting a higher reabsorption of collagen, which were substantial differences between responders and non-responders. These results point to the advantageous impact of RDN effect on CV fibrosis in patients presenting hypertensive cardiac impairment and target-organ harm.

Jiang et al. reported that RDN efficacy on decreasing ventricular arrhythmias triggering is not inferior compared to Metoprolol, in rats presenting MI (Figure 5). The pathway could be correlated to cardiac fibrosis decreasing, Cx43 expression controlling and sympathetic nerves remodelling [97]. In dogs, sympathetic stimulation of renal nerves for 3 h enhanced LSG neuronal activity, and make easy the occurrence of ventricular arrhythmias in the course of acute myocardial ischemia. Remarkably, the increase in VAs could be blunted by LSG ablation. Also, RDN has reduced ventricular ectopic activity and ventricular fibrillation [98] in induced-acute ischemia and reperfusion [99] (Figure 6) or induced-MI by a permanent coronary occlusion [98] in pig models [85,100,101]. Moreover, in a HF dog model induced by ventricular high-rate pacing, RDN lessened the ventricular remodeling progression [102,103]. The number of natural ectopic ventricular beats and the consequent ventricular dysfunction could be inhibited by RDN modulating the SNS [85]. Likewise, arrhythmogenic elongation of QT-interval caused by mimicked sleep apnea [104] or by cesium [105] could be lightened by RDN.

Experimental data demonstrated that RDN decreases the SNS activation as measured by reduced renal concentrations of catecholamine. Moreover, RDN remarkably improved LV longitudinal strain, reduced the end-systolic volume, and lessened cardiac fibrosis, leading to an enhanced cardiac function. Fascinatingly, RDN lowered neprilysin activity, increasing cardioprotective B-type natriuretic peptide (BNP) concentrations [107]. Usually in congestive HF, cardiac volume overload provokes releasing of A-type natriuretic peptide and BNP, which has both diuretic effects and cardioprotective properties [108,109]. Boosted neprilysin production, which enzymatically degrades theoretically protective natriuretic and other bioactive peptides, was found to participate in the progress of LV dysfunction. The raise of neprilysin is correlated to bad outcomes. Its inhibition with sacubitril/valsartan offers extraordinary advantages in HF individuals [110] (Figure 7).

Circulating NA concentrations predict mortality in HF individuals presenting low LVEF [112]. Unusually, either cardiac or renal NA spillover are increased in all chronic HF stages matched to healthy subjects. However, there are differences between the absolute renal and cardiac NA spillover levels (~25% vs. ~3%, respectively), which suggests that the kidney is crucial to total NA spillover in HF [113,114].

A small case series has shown that, in individuals with dilated cardiomyopathy and an electrical storm, RDN was capable of lowering the rate of ICD shocks and ventricular ectopic activity [115]. Several other case series [116,117] and an international multicentre registry also have demonstrated these anti-arrhythmic effects [118]. RDN may be mainly advantageous for HF subjects presenting with refractory arrhythmias that are not able to put up with maximal β-blocker dosage and are not appropriate for VT ablation. Otherwise, RDN can be an adjunct strategy in patients undergoing catheter ablation. In HF subjects, RDN fell NT-pro BNP concentrations and was harmless, showing no worsening of other indices of cardiac and renal function [119]. Another study reported a reduction in the rate of malignant VAs and proper ICD therapies in advanced stages of chronic kidney disease (CKD) and HF patients [120].

Patients with advanced HF have present a huge ANS imbalance, which per se has a substantial impact on the prompting and maintenance of VAs [121,122]. The sympathetic overdrive on the heart can aggravate the hazardous already present settings, such as ischemia, dilated cardiomyopathies, or underlying rhythm irregularities to trigger life-threatening arrhythmias [121,123]. In this context, the intrinsic cardiac nervous system is a protagonist regarding atrial and ventricular function regulation, as the heart receive both sympathetic and parasympathetic inputs [124]. Hence, comprehending the role that the ANS exerts in the pathogenesis of arrhythmias and how it can be blunted or blocked may provide essential clues for either prevention or treatment of VAs [125,126] (Figure 8).

Recently, Tsai and colleagues showed in ambulatory canines that bilateral RDN, possibly through afferent renal innervation interruption, led to the substantial brain stem and bilateral stellate ganglion remodeling at eight weeks post-procedure [85]. These changes were associated with reduced 18FDG- uptake in the brainstem, left stellate ganglion nerve activity and atrial tachyarrhythmia events. The authors concluded that neural remodeling in the brain stem and stellate ganglion may partially explain the described antiarrhythmic effects of RDN [85].

Trans-synaptic degeneration is a phenomenon in the central and peripheral nervous system that may remain active both at the level of the insult and in remote brain structures for as long as one year after trauma [127]. These progressive alterations may underlie some of the long-term functional consequences after the initial injury (i.e., RDN) as shown in Figure 9, which summarizes the various direct and indirect connections between renal sympathetic nerves and the stellate ganglion. Meckler and colleagues showed that approximately 10% of renal sympathetic neurons in cats originated from the thoracic chain ganglia [85]. Given the connections between these two structures, RDN may directly result in retrograde cell death of the stellate ganglion. Furthermore, the application of fluorescent dyes in the renal nerves results in fluorescent labeling of the sympathetic cell bodies in paravertebral and prevertebral ganglia [128,129,130].

Since the sympathetic preganglionic neurons that project to the stellate ganglion are dispersed in spinal cord segments T1-T10 [131], there is more than enough chance to interconnect with the preganglionic cells that link indirectly with sympathetic nerve fibers surrounding the renal arteries. Nonetheless, it is probable that some other pathways participate to the trans-synaptic degeneration [85] since the ganglion cells of renal afferent nerves located in thoracic and lumbar spine dorsal root ganglia also link to the posterior and lateral hypothalamic nuclei and the locus ceruleus [132,133]. Overall, these findings indicate that lasting effects of RDN may be mediated by remodeling of critical brainstem areas and the stellate ganglia. Theoretically, VAs and consequently episodes of SCD could be perfectly suppressed by the action of RDN in this mechanism.

### Cardiac Conduction System as a Target for RDN

The cardiac conduction system is composed of the sinoatrial node (SAN), the atrioventricular node (AVN), the atrioventricular (AV) bundle (bundle of His) and its branches, as well as the Purkinje cells.

The normal electrical stimulus originates in the SAN, which pursues the highest depolarization intrinsic rates, and therefore has a chronotropic function, working as a pacemaker. This electrical impulse is called sinus rhythm disperses from its origination in the SAN to the atrial cells and the AVN. The duration of conduction between these two nodes is ~50 ms. Also, the Bachmann’s bundle, a specific band bundle pathway between the right and left atrium (RA and LA, respectively) leads the impulse straight from the former to the latter. Irrespective of the path, as the impulse achieves the AV septum, the cardiac skeleton connective tissue avoids the electrical wave from dispersal into the ventricular myocytes. This wave of depolarization prompts muscular contraction, which commences in the RA, and the goes through the upper portions of both atrial chambers, and posteriorly downwards via the contractile cells. Then, these cells initiate contraction (inotropy) from the atrial top to the inferior segments, effectively propelling blood into ventricular chambers. Afterward, the electrical stimulus dispersal straight to ventricles and a critical pause (~100 ms) occurs before the AVN depolarizes and conveys the impulse to the bundle of His (AV bundle). This halt is crucial to the normal heart performance, once it permits the atrial cells contraction conclusion, pumping the blood into the ventricular chambers prior to the stimulus being transmitted to the ventricular myocytes.

When the SAN stimulates the AVN at huge high frequencies, the AVN may conduct impulses up to 220 bpm, which determines the usual maximum heart rate (HR) in healthy subjects. The hypersympathetic state existents in HF provokes non-effective contraction at a HR and therefore is associated with progressive myocardial remodeling, deterioration of LV function, and worsening symptoms [112,134]. The “Systolic Heart Failure Treatment With the If Inhibitor Ivabradine Trial” (SHIFT) [27] demonstrated that Ivabradine when matched to control-placebo in subjects with HF and HR >70 bpm, in spite of optimal medical therapy, was correlated with better results, established as CV death or hospital admittance owing to HF. The SHIFT Trial confirmed the significant role of HR in the pathophysiology of HF. Altered SAN automaticity is translated by modifications of the HR, which is used to quantify cardiac autonomic modulations through several methods, such as heart rate variability (HRV), baroreflex sensitivity (BRS), HR turbulence (HRT), HR deceleration capacity (HRDC) and T wave-alternans (TWA).

Heart rate variability: Both sympathetic and parasympathetic nervous systems modulate SAN automaticity. Modulation of HR by respiration is recognized event mediated by cardiopulmonary afferent inputs and central interactions between CV and respiratory networks [135,136]. Modifications of the HR are easily assessed through ECG and is used to calculate cardiac autonomic modulations, such as HRV. Many different approaches can measure HRV. The most usual is frequency or time domain analysis.

Baroreflex sensitivity: BRS is a marker of autonomic input to the SAN and measured by the reflex alterations in R-R interval in reaction to provoked changes in BP. Often, it is estimated by characterizing the extent of induced bradycardia in response to phenylephrine. BRS reduces with age progression and is reduced in hypertensive or HF individuals [137,138]. The ATRAMI study demonstrated that, post-MI, the SD of the average of a normal sinus to a normal sinus interval (SDANN) < 70 ms or BRS < 3.0 ms/mmHg with LVEF <35% supported a substantial risk of cardiac mortality [138]. Although, daily physical activity avoids VF triggered by acute MI through either lowered sympathetic or heightened parasympathetic tone [139].

Heart rate turbulence: HRT is an index of modifications in sinus rate post to a ventricular ectopic beat followed by a compensatory pause. Typically, the sinus rate primarily accelerates and slows subsequently, but this event is disrupted in numerous cardiac diseases. Anomalous HRT is related to increased total mortality and SCD in patients presenting coronary artery disease and dilated cardiomyopathy [140]. Also, the relative risk for HRT abnormal values is a strong predictor of mortality [141].

Heart rate deceleration capacity: HRDC is based on a signal processing algorithm to separately distinguish HR deceleration and acceleration, which in turn differentiate between vagal and sympathetic elements. HRDC is supposed to be a better predictor of mortality post-MI than LVEF and SDANN [142].

T wave alternans: TWA is beat-to-beat variability in T-waves amplitude or morphology. TWA replicates ventricular repolarization temporal heterogeneity or dispersion. TWA was primarily used as a tool for SCD risk stratification in individuals presenting ischemic and nonischemic cardiac diseases. The negative predictive value of this approach is high, and a negative test substantially predicts the absence of VT and VF risk [143].

The SAN function is hugely dependent on the sympathetic neurons (SNs) control. In resting conditions, even in sinus rhythm, physiological HRV can be noticed, without significant change in catecholamine concentration in the plasma compartment. While the neurogenic mechanisms controlling HRV have primarily been credited to the parasympathetic neurons (PSNs), and therefore to vagal influence, the SNs contribute through some other aspects (e.g., hormones, respiratory rate, hemodynamic reflexes, and temperature) for the HRV modulation [144]. Reliably, neuronal input is blocked to the heart by either atropine or β-blockers, and a heart transplant, provoking cardiac denervation, ablate such variability, leading to a static HR [85,145,146]. At the molecular level, the sympathetic input speeds up SAN automaticity. The SAN innervation and the accuracy of the chronotropic control are indirect hints that validates the theory that direct neurocardiac coupling causes neurogenic regulation of the cardiac pacemaker function. Hence, RDN may indirectly impact on the cardiac chronotropism via afferent renal nerves disruption, as previously explained, which would benefit HF patients [27] and at certain extent would prevent the worsening of LV dysfunction, which by its turn would decrease the odds of VAs and SCD occurrence [147].

The bundle of His (AV bundle) comes from the AVN and continues over the interventricular septum prior to separating in the left and right bundle branches (LBB and RBB, respectively). RBB portions are located in the moderator band and supply the right papillary muscles. Due to this linking, each papillary muscle gets the electrical wave almost simultaneously, beginning to contract at the same time, before the rest of the ventricular myocytes. Both RBB and LBB run down and achieve the cardiac apex (~25 ms) where they link up with the Purkinje fibers. The AV period elongation delays the systolic contraction, which might impact on initial diastolic filling [148]. Once a delay in ventricular contraction occurs, LV diastolic pressures will overdo atrial pressure and diastolic mitral regurgitation will follow. Then, the LV pre-load lost leads to a reduction in its contractility, due to loss of the Starling mechanism. Both inter- and intra-ventricular conduction delays lead to ventricular dyssynchrony, prejudicing the cardiac performance and reducing stroke volume and systolic BP. Uncoordinated papillary muscle function may prompt or aggravate functional systolic mitral regurgitation, while an impaired performance promotes adverse LV remodeling. CRT helps to restore AV, inter- and intra-ventricular synchrony, improving LV function, reducing functional mitral regurgitation and inducing LV to reverse remodeling, as demonstrated by increases in LV filling time and LVEF, and decreases in LV end-diastolic- and end-systolic volumes, mitral regurgitation and septal dyskinesis [149,150,151]. The primary mechanism of benefit is likely to vary from one patient to the next and within an individual patient over time. As previously mentioned CRT implantation resulted in a significant reduction of MSNA in patients with HF [50].

The Purkinje fibers disperse the impulse to the ventricular myocardial contractile cells. They lengthen all over the myocardium from the apex toward the AV septum and the cardiac base. The Purkinje fibers pursue a rapid intrinsic conveyance frequency, and thus the impulse takes ~75 ms to spread over all the ventricular myocytes. As the electrical wave starts at the apex, the contraction also initiates at the apex, traveling toward the cardiac base, allowing the blood to be pushed out from the ventricles to pulmonary artery trunk and aorta. However, as aforementioned, this function is severely impaired in advanced systolic HF.

Sympathetic nerve sprouting and disturbed innervation are relevant in cardiac arrhythmia scenarios [152]. Sema3a works as a convincing neural chemorepellant, controlling axon/dendrite growing and neuronal migration [153]. Fascinatingly, Sema3a is strongly expressed in the developing heart, and its expression progressively falls with development [154]. Also, Sema3a seems to be a negative controller of cardiac sympathetic innervation, acting in its modeling by inhibiting neural growth. Its overexpression has led to sustained VAs in mice (Figure 10), upregulation of β-ARs density and extended action potential time, which shows the relevance of neurotrophic factors in promoting neuronal survival, innervation patterning and in arrhythmia pathogenesis [126].

Disturbance of the ANS balance, comprising hypersympathetic drive, has been associated with VAs pathogenesis. Sympathetic overactivity has been reported to come ~30 min first to the onset of VAs in humans [155]. Spontaneous sympathetic nerve discharge from the LSG in a canine model of SCD immediately triggered malignant VAs and death [156]. Nerve sprouting is correlated with VT/VF and SCD as well. Subjects presenting VT/VF history had increased sympathetic nerve sprouting, mostly in the margin of the normal myocardium and scar tissue, in comparison to the ones showing similar structural cardiac disease without arrhythmias [157]. Likewise, ^131^I MIBG scans from these patients proposed that heterogeneity of sympathetic innervation associates with VAs risk [59]. Pre-clinical data have demonstrated that rabbits on a high cholesterol diet developed myocardial hypertrophy and sympathetic hyperinnervation without coronary artery disease and had a higher incidence of VF [158]. Thus, even in the presence of heterogeneous sympathetic nerve sprouting or hyperinnervation, the numbness of the SNS and the reduction of the over-sympathetic discharges arriving at the heart, through afferent renal nerves ablation can potentially lessen the VT/VF and SCD rates.

## 3. SNS Management to Diagnose, Prevent and Treat HF, VAs, and SCD: Challenges and Future Directions

SNS management has emerged as a useful tool for diagnosing, preventing and treating HF, VAs, and SCD [59,120,121]. However, its configuration and clinical manifestations and disturbances are numerous and multifaceted. Both clinical history and existing exams may not be enough for uncovering the underlying sources, and advanced tests may potentially take place, such as [^11^C]-HED PET cardiac imaging in HF patients [59]. Once autonomic assessments have not followed standard patterns in different populations, these characterize an obstacle to implementing such diagnostic tools. Also, pediatric patients cannot undergo several autonomic function tests. In both infants and young adults, data about the efficacy of the methods are limited. The SCD mechanisms in these populations are manifold, mostly occurring without heart failure development, and comprising general etiologic categories including heritable and acquired cardiomyopathies and arrhythmia syndromes (channelopathies, which depend on genetic tests), structural congenital heart diseases, myocarditis and coronary abnormalities. In these cases, non-invasive quantitative tests that require minimal participation and cooperation have been used. The proportion of the detected vs. undetected risk of SCD varies by diagnosis, as does our ability to mitigate the risk of cardiac arrest by prophylactic therapy and other preventative measures. These factors strongly affect the utility of diagnostic screening in asymptomatic individuals. This evidence gives us a reasonable direction for future investigations, focusing on specific risk prediction according to the underlying disorder.

Broad and detailed explorations of how RDN decreases the burden of HF, VAs and SCD are immediately necessary. So far, the majority of data published on this purpose is limited to animal studies [100], and initial studies in humans have small cohorts, have been performed in HF patients, and are non-randomized, unblind and non-controlled [118]. Moreover, the exact mechanisms through RDN seem to reduce the burden of VAs in HF, and therefore prevent SCD events, are unknown. As RDN has had a beneficial impact on the most critical comorbidities, such as atrial fibrillation [59] and diabetes [120], prospective clinical trials are required to analyze the probable RDN advantageous effects of RDN in HF.

## Figures and Tables

**Figure 1 ijms-20-02430-f001:**
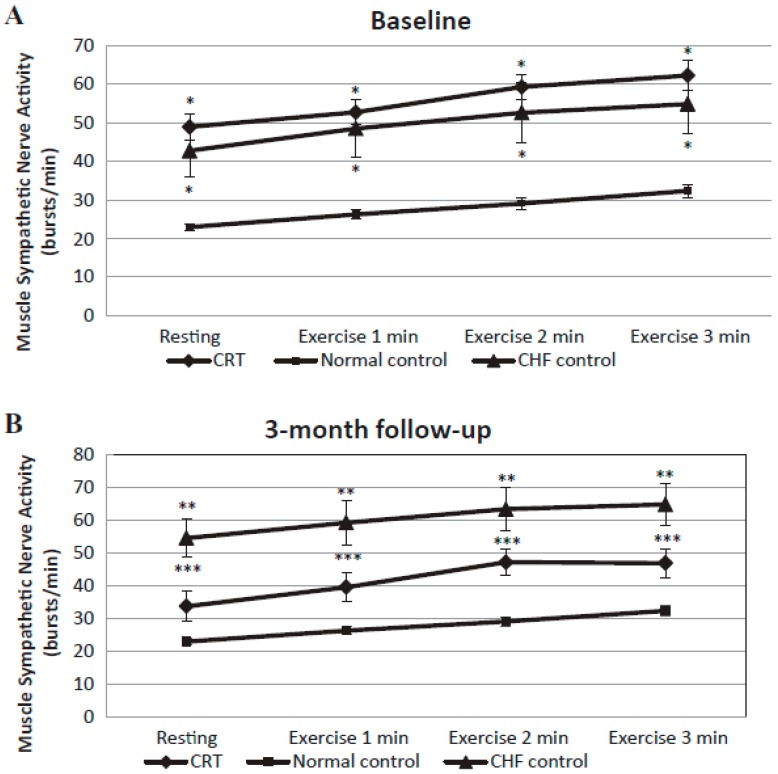
Muscle sympathetic nerve responses during moderate static exercise in congestive heart failure patients (CRT and control) compared with normal control subjects. (**A**) Moderate static handgrip exercise at baseline. (**B**) Moderate static handgrip exercise after a 3-month follow-up. * *p* < 0.001 *vs.* normal control. ** *p* = 0.005 *vs.* normal control. *** *p* = 0.003 *vs.* normal control. CHF = congestive heart failure, CRT = cardiac resynchronization therapy. The bars represent the standard deviation [50].

**Figure 2 ijms-20-02430-f002:**
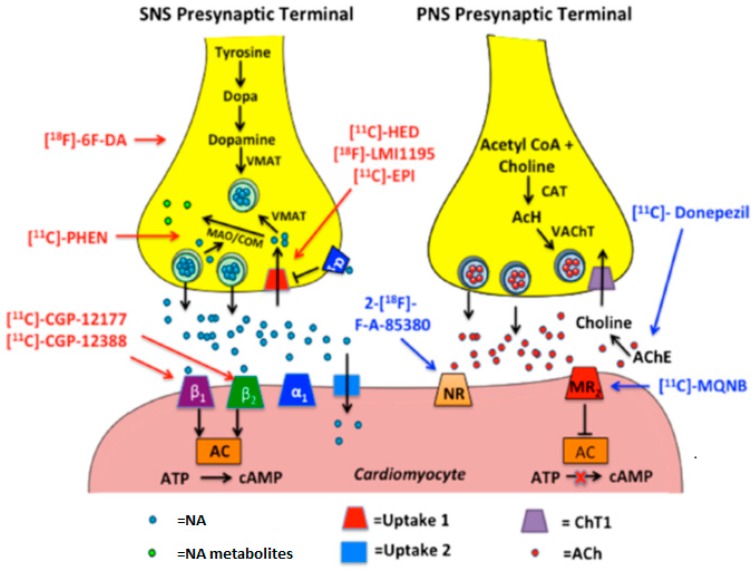
The illustration depicts postganglionic SNS and PNS nerve endings. The left panel displays the synthesis and release of noradrenaline in postganglionic SNS nerve endings and subsequent binding to postsynaptic receptors on cardiomyocytes. The tracers in red depict SNS pre- and post-synaptic radio analogs. The right panel shows the synthesis and release of acetylcholine in the terminal nerve ending and varicosities of postganglionic PNS nerve endings and subsequent binding to postsynaptic receptors on cardiomyocytes. Tracers in blue depict PNS pre- and post-synaptic radio analogs. AC = adenylyl cyclase, ACh = acetylcholine, AChE = acetylcholinesterase, ATP = adenosine triphosphate, CAT = choline-acetyl-transferase, COM = catechol-O-methyltransferase, cAMP = cyclic adenosine monophosphate, MAO = monoamine oxidase, MR2 = muscarinic receptor 2, NA = noradrenaline, NR α4β2 = nicotinic receptor, VMAT = vesicular monoamine transporter, 18F-6F-DA 6-18F-fluorodopamine, PHEN =phenylephrine, EPI = epinephrine (adrenaline), HED = hydroxyephedrine, MQNB = (R,S)-N-[^11^C]-methyl-quinuclidin-3-yl benzilate [56].

**Figure 3 ijms-20-02430-f003:**
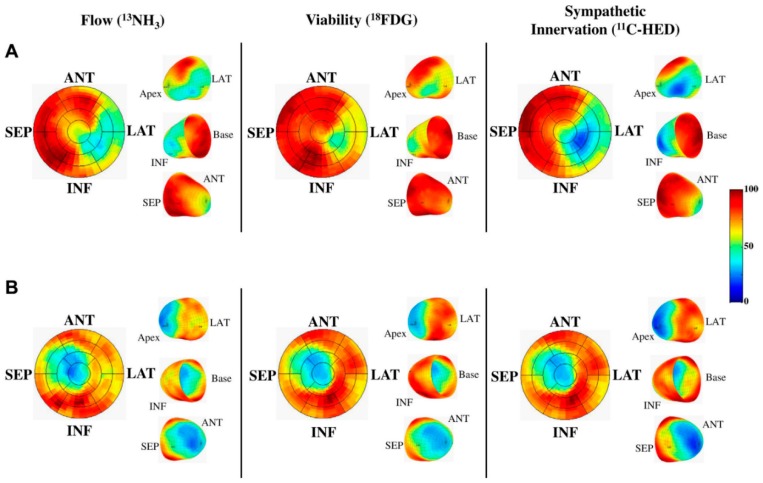
PET imaging of flow, viability and sympathetic innervation. (**A**) A subject experiencing sudden cardiac arrest (SCA). There is a mismatch in infarct size (reduced ^18^F-2-deoxyglucose [^18^FDG]), which was smaller than the volume of sympathetic denervation (reduced ^11^C-meta-hydroxyephedrine [^11^C-HED]). There was also reduced perfusion (^13^N-ammonia [^13^NH_3_]) with preserved ^18^FDG indicating hibernating myocardium. In contrast, (**B**) shows a subject with matched reductions in flow, infarct volume, and sympathetic denervation. ANT = anterior; INF = inferior; LAT = lateral; PET = positron emission tomography; SEP = septum [59].

**Figure 4 ijms-20-02430-f004:**
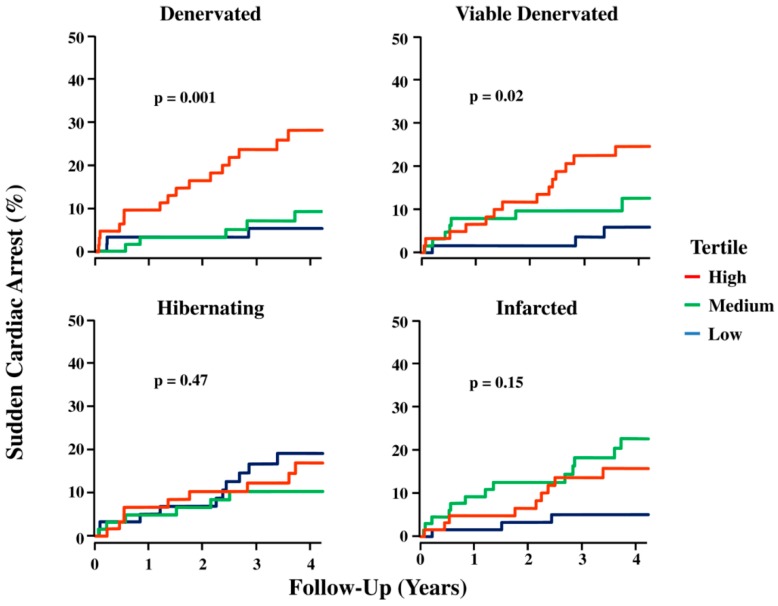
PET parameters and sudden cardiac arrest. Kaplan–Meier curves show the incidence of sudden cardiac arrest for tertiles of PET-defined myocardial substrates (median follow-up 4.1 years). As continuous variables, the total volume of denervated myocardium, as well as viable denervated myocardium, predicted sudden cardiac arrest. Neither infarct volume nor hibernating myocardium was significant as continuous variables. SCA = sudden cardiac arrest [59].

**Figure 5 ijms-20-02430-f005:**
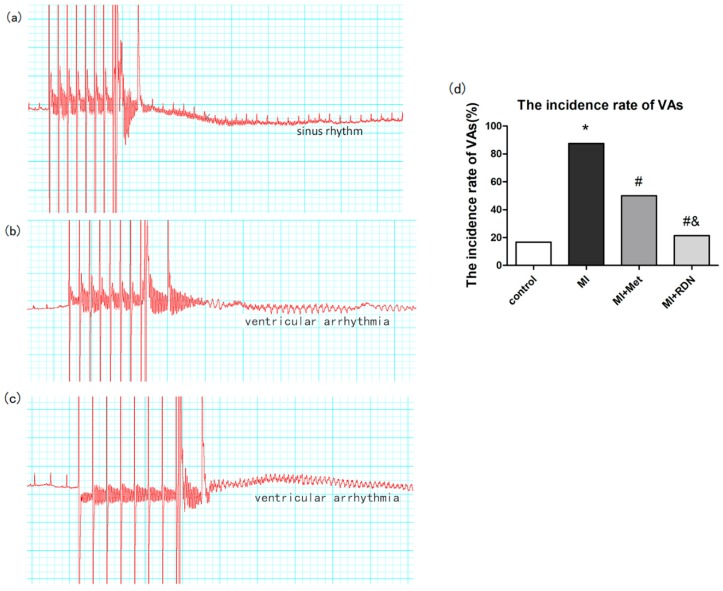
RDN significantly reduced the incidence of ventricular arrhythmias. Representative ECG of electrical stimulation, including sinus rhythm (**a**), ventricular arrhythmias (**b**) and (**c**). Ventricular arrhythmias were less easily induced in RDN group rather than in MI group and Met group (**d**). (* *p* < 0.05 vs. Control group; # *p* < 0.05 vs. MI group; & *p* < 0.05 vs. Met group). MET = metoprolol. MI = myocardial infarction. RDN = renal denervation [97].

**Figure 6 ijms-20-02430-f006:**
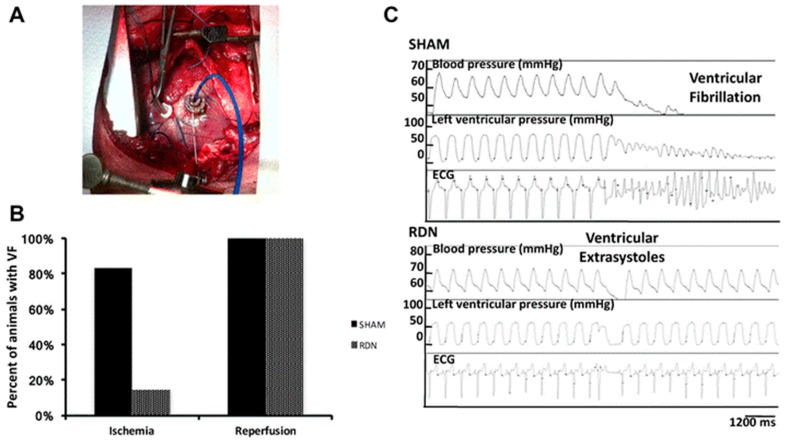
Effects of renal denervation on ventricular fibrillation in a pig model for ventricular ischemia and reperfusion. (**A**) Representative view of the left ventricular during ischemia reperfusion experiments. Atrial electrophysiology was recorded by an epicardial catheter. (**B**) Incidence of VF during ischemia and the reperfusion phase in RDN-treated compared to SHAM-treated pigs. (**C**) Representative hemodynamics and electrocardiographic (ECG) tracings during 20 min of left anterior descending coronary artery ligation followed by reperfusion in a SHAM-treated and a RDN-treated animal. RDN = renal denervation. VF = ventricular fibrillation [99,106].

**Figure 7 ijms-20-02430-f007:**
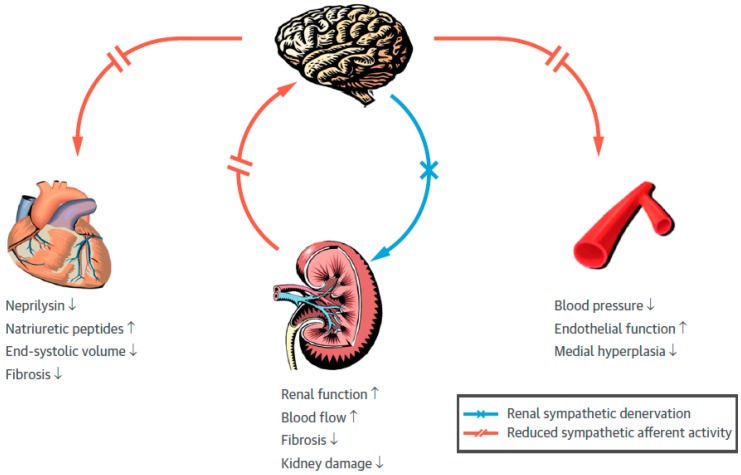
Effects of renal sympathetic denervation. Sympathetic efferent activation is generated in the central nervous system. Efferent sympathetic nerves (red lines) target the heart, kidney, and vessels and produce neprilysin activation (new finding), inotropic effects, but also fibrosis, beta adrenergic down-regulation and contractile dysfunction. In the kidney, the vasoconstriction increases renin activation and enhanced sodium water retention, contributing to fibrosis and kidney damage. Peripheral arterial afterload increases by vasoconstriction and, in the long run, BP is increased. Atherosclerosis is initiated, and media hyperplasia takes place. Interruption of afferent as well as efferent kidney nerve fibers by using renal sympathetic denervation (red cross) reduces sympathetic outflow (interrupted red lines) and potentially reverses these pathological findings [111]. Down arrows = efferent sympathetic activation. Up arrows = afferent sympathetic activation.

**Figure 8 ijms-20-02430-f008:**
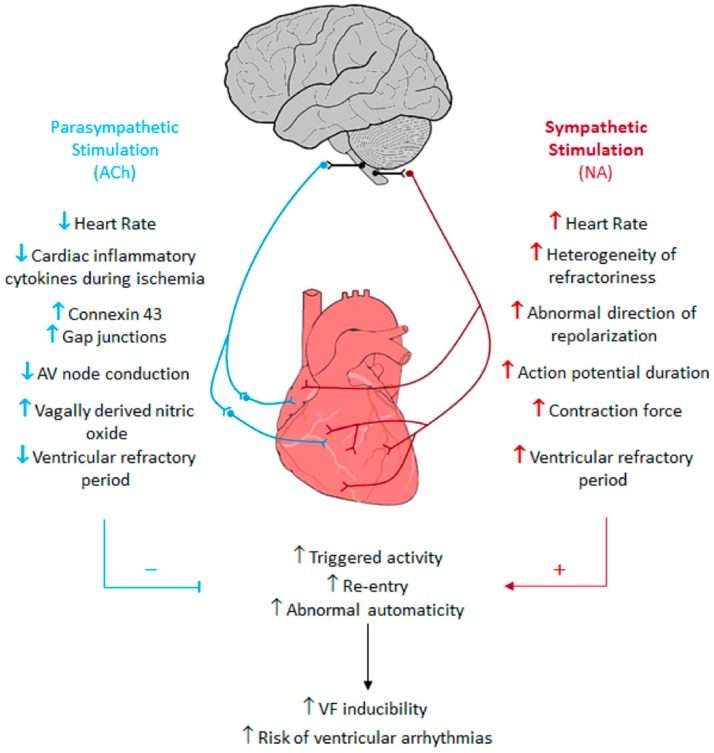
Cardiac arrhythmogenic effects provoked by the sympathetic and parasympathetic nervous systems [126]. Blue = parasympathetic nervous system. Red = sympathetic nervous system. Black = Connections between the brain and the parasympathetic and sympathetic nervous systems. Up arrows = increase. Down arrows = decrease. Red line and + = provoke these events. Blue line and − = inhibits these events.

**Figure 9 ijms-20-02430-f009:**
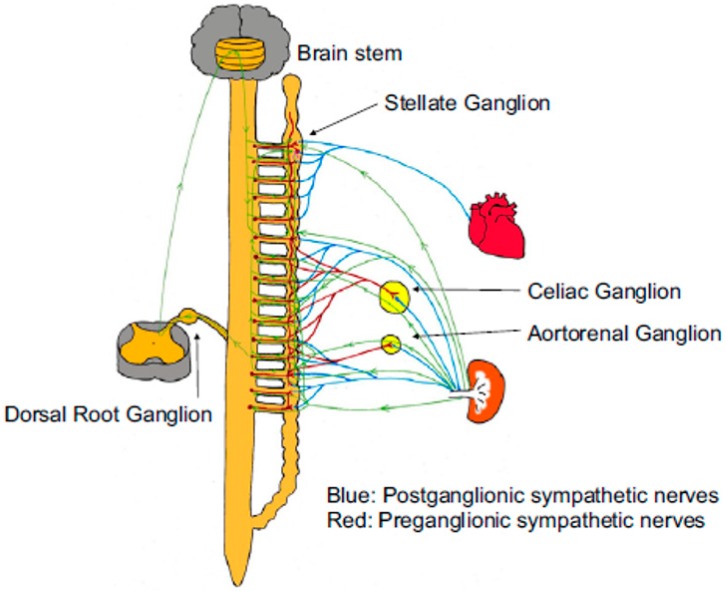
Representations of possible connections amongst different nerve structures [85].

**Figure 10 ijms-20-02430-f010:**
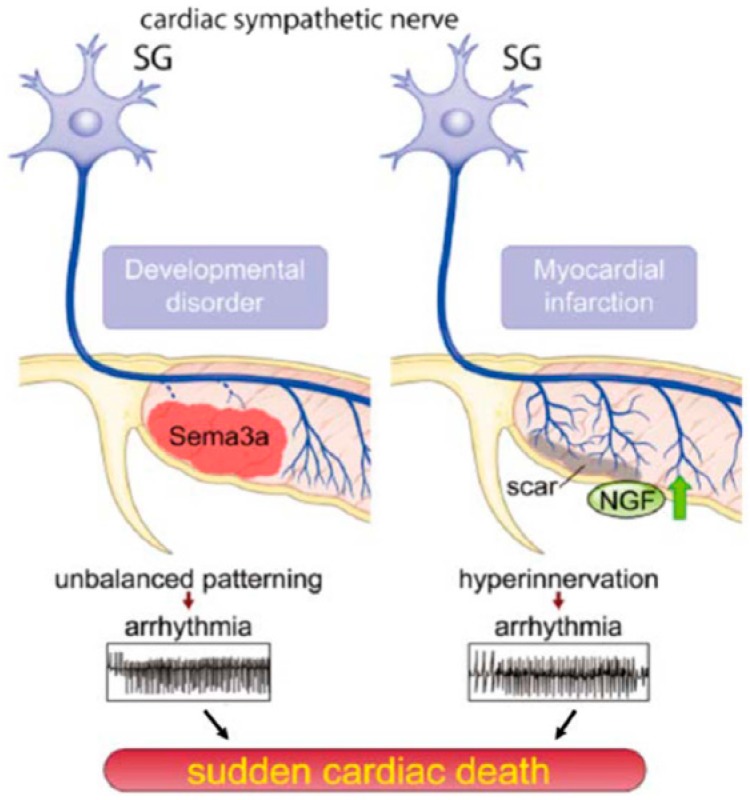
Cardiac innervation patterns provoking sudden cardiac death. Left: Overexpression or underexpression of Sema3a, a class 3-secreted semaphorin, which acts as a potent neural chemorepellant, leads to abnormalities in innervation patterning of sympathetic nerves causing ventricular arrhythmias and sudden cardiac death. Right: Overexpression of NGF leads to a disrupted patterning of sympathetic neurons leading to hyperinnervation, ventricular arrhythmias, and sudden death [121]. Upregulation of secreted nerve growth factor (NGF) from cardiomyocytes in diseased heart (NGF and green arrow) may cause lethal arrhythmia (red arrows) and SCD (black arrows).

**Table 1 ijms-20-02430-t001:** Significant predictors of time to SCA.

Variable	Univariate		Multivariate	
	HR (95% CI)	*p*-Value	HR (95% CI)	*p*-Value
Denervated myocardium	1.057 (1.023–1.092)	0.001	1.069 (1.023–1.117)	0.003
Viable denervated myocardium	1.067 (1.008–1.130)	0.025		
Infarcted myocardium	1.029 (0.990–1.069)	0.15		
Hibernating myocardium	0.950 (0.822–1.099)	0.49		

CI = confidence interval; HR = hazard ratio; PET = positron emission tomography; SCA = sudden cardiac arrest [59].

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
