# Peer review of "New Approaches in the Management of Sudden Cardiac Death in Patients with Heart Failure—Targeting the Sympathetic Nervous System"

_ijms, 2019, doi:10.3390/ijms20102430_

Reviewer 1 Report

The authors performed a review focused on novel approaches in SCD prevention.

The review is well-written and all data is explained in detail.

Only minor points should be improved:

1.- what about young population at risk of SCD?. The methods reported are also effective for this cohort of young-adult people?

2.- concerning methods in the management, any disavantatge? Any secondary effect reported in RDN?

3.- Authors focused on CVD in general, mainly in HF. The management will be the same in other pathologies associted with SCD, such as familial cardiomiopathies?

Author Response

We have addressed every reviewer comment, point by point, as carefully as possible. All changes are marked in yellow. The manuscript was thoroughly revised. We hope that the paper now looks good enough to be accepted in this reputable journal.

Please find attached the file.

We do appreciate your kind attention.

Reviewer 2 Report

In the study titled "New approaches in the management of sudden cardiac death - targeting the sympathetic nervous system", Kuichi MG et al., review literature regarding the pathophysiology of sudden cardiac death (SCD). A particular focus is on the role of sympathetic nervous system (SNS) in the progression of SCD. 

Comments to the manuscript are;

The manuscript is largely focused on the clinical data and does not take into account the molecular or cellular effects of SCD and SNS manipulation/renal denervation. 

Also, it would be helpful to the reader to have an illustration summarizing potential mechanisms involved relating SNS to SCD pathophysiology.

Another interesting area would be have a section in the review targeting the affect of SNS manipulation on the different cell types of the heart and its impact on the SCD 

Authors should provide a limitations/challenges and future directions section at the end to highlight the current problems and provide the reader food for thought going forward in the regards to the role of SNS in the management of SCD.

Author Response

We have addressed every reviewer comment, point by point, as carefully as possible. All changes are marked in yellow. The manuscript was thoroughly revised. We hope that the paper now looks good enough to be accepted in this reputable journal.

Please find attached the file.

We do appreciate your kind attention.

Round  2

Reviewer 2 Report

All concerns have been addressed. No further comments